# Bet hedging in a unicellular microalga

Si Tang[1], Yaqing Liu [1], Jianming Zhu[1], Xueyu Cheng[1], Lu Liu[1], Katrin Hammerschmidt [2] ✉, Jin Zhou[1] ✉ & Zhonghua Cai [1,3] ✉

Understanding how organisms have adapted to persist in unpredictable environments is a fundamental goal in biology. Bet hedging, an evolutionary adaptation observed from microbes to humans, facilitates reproduction and population persistence in randomly fluctuating environments. Despite its prevalence, empirical evidence in microalgae, crucial primary producers and carbon sinks, is lacking. Here, we report a bet-hedging strategy in the unicellular microalga *Haematococcus pluvialis*. We show that isogenic populations reversibly diversify into heterophenotypic mobile and non-mobile cells independently of environmental conditions, likely driven by stochastic gene expression. Mobile cells grow faster but are stress-sensitive, while non-mobile cells prioritise stress resistance over growth. This is due to shifts from growth-promoting activities (cell division, photosynthesis) to resilience-promoting processes (thickened cell wall, cell enlargement, aggregation, accumulation of antioxidant and energy-storing compounds). Our results provide empirical evidence for bet hedging in a microalga, indicating the potential for adaptation to current and future environmental conditions and consequently conservation of ecosystem functions.

Organisms must adapt to changing environments. While most populations can accommodate regular environmental changes, such as daily diurnal and annual seasonal cycles, dealing with random changes or fluctuations poses a challenge. Today, the environment is currently changing at an unforeseen high rate, mainly due to anthropogenic activities, generating novel and unexpected environmental fluctuations that pose urgent challenges to every living organism[1,2]. Therefore, the question of whether and how organisms adapt to unpredictable environments is crucial.

In evolutionary theory, bet hedging is proposed as a successful adaptive strategy to cope with randomly fluctuating environments[3]. In unicellular organisms, bet hedging means that an isogenic population can increase its long-term fitness, at the expense of lower short-term fitness, through the stochastic development of different phenotypes[4–6]. Different phenotypes may be well-adapted to different environmental conditions, with individuals with each phenotype exhibiting varying levels of reproductive success and survival rates that differ based on external conditions. Unlike real-time sensing and

subsequent response to change, bet hedging is advantageous because it enables a subpopulation to display the phenotype that will be adaptive in a future environment prior to any environmental changes. Upon sudden environmental changes, a bet-hedging strategy offers a rapid and reliable mechanism for evading extinction[7].

Bet hedging has attracted growing interest and has been reported in diverse research areas[6,8–13], both theoretically and experimentally, as more evidence is emerging from various model organisms. To our knowledge, however, there has been no direct evidence for bet hedging in microalgae, which are among the most important primary producers on Earth and are of critical ecological importance, participating in global biogeochemical carbon and nitrogen cycles[14,15]. As their physiology and survival are severely affected by global climate change[16], we urgently need to increase our understanding on how they adapt to rapidly changing environments.

Here, the genetically clonal microalgal species *Haematococcus pluvialis* is used as a model organism to study bet hedging in microalga. *H. pluvialis* (*Chlorophyceae, Volvocales*) is a widely distributed,

[1]Shenzhen Public Platform for Screening and Application of Marine Microbial Resources, Tsinghua Shenzhen International Graduate School, Shenzhen 518055 Guangdong Province, PR China. [2]Institute of General Microbiology, Kiel University, Kiel, Germany. [3]Technology Innovation Center for Marine Ecology and Human Factor Assessment of Natural Resources Ministry, Tsinghua Shenzhen International Graduate School, Shenzhen 518055 Guangdong Province, PR China. ✉e-mail: katrinhammerschmidt@googlemail.com; zhou.jin@sz.tsinghua.edu.cn; caizh@sz.tsinghua.edu.cn

buoyant unicellular biflagellate freshwater microalga. Thanks to the commercial value of the species, known as the best natural source for the bio-product astaxanthin, its morphological and physiological changes as well as metabolic characteristics have been extensively studied in different cultural environments[17–19]. A particularly interesting phenomenon that has been repeatedly reported[17,18,20] and that we have also observed in both culture flasks and bioreactors is the coexistence of two behaviourally distinct subpopulations, namely mobile and non-mobile green cells.

Until now, it has been assumed that external stress factors induce the transformation of mobile to non-mobile green cells, which then turn into dormant red cells when conditions deteriorate further[19]. However, the coexistence of mobile and non-mobile subpopulations in the green stage remains unexplored, and the ecological and functional significance of this phenotypic heterogeneity is not understood.

As non-mobile cells appear earlier than expected in standard culture conditions, we doubt that external stress factors are the sole trigger for the observed phenotypic heterogeneity. Instead, we propose here that the coexistence of two phenotypes in the *H. pluvialis* populations can be explained by bet hedging (see Fig. 1a for a comparison of the stress-induced cellular transformation (traditional view), i.e., the accumulation of carotenoids accompanied by metabolic and morphological changes, termed the carotenogenic response[21]). In the case of bet hedging, we expect the mobile subset of the population to produce more offspring but be more susceptible to external stressors, while the non-mobile subgroup should be more stress resistant at the expense of lower reproduction.

In this study, we used laboratory populations of *H. pluvialis* to test the following: (1) Do the two subpopulations of *H. pluvialis* behave differently under standard and stress conditions? This would be expected if they have different stress resistance. (2) Does a hetero-phenotypic population show lower variance in reproductive fitness under varying environments as compared to a phenotypically uniform population? This is a canonical feature of bet-hedging populations[3,22]. To this end, we combined growth and survival experiments with microscopic, physiological and transcriptomic analyses.

Here we find that isogenic populations of *H. pluvialis* diversify into mobile and non-mobile cells regardless of environmental conditions, likely due to stochastic gene expression. Mobile cells grow faster but are more sensitive to stress, while non-mobile cells prioritize stress resistance over growth. This shift is a result of a transition from growth-promoting activities, such as cell division and photosynthesis, to resilience-promoting processes, such as thickening of the cell wall, cell enlargement, aggregation, and accumulation of antioxidant and energy-storing compounds. Measurements of population growth and survival rates show that heterophenotypic populations have reduced variance in fitness under different conditions, complying with the bet hedging theory. This study provides an experimental report of bet-hedging behaviour in the phyla of microalgae. It provides fundamental insights into how these organisms cope with unpredictable environments and offers a unique perspective on managing future species or ecosystems.

## Results

### Phenotypic diversification of *H. pluvialis*

First, a population starting with 100% mobile *H. pluvialis* cells was grown in fresh BBM medium without external stress. After 24 h, the differentiation into a hetero-phenotypic population, composed of mobile and non-mobile cells, could be detected (Fig. 1b, c). As the mobile cells converted into non-mobile cells, the ratio of mobile cells decreased, while the percentage of non-mobile cells increased over time. Specifically, the percentage of mobile cells decreased from 100% to 8.8%, while the non-mobile cells accounted for 91.2% of the total population at the end of the experiment (Day 20, Fig.1c). It is interesting to note the early occurrence of the non-mobile phenotype,

which already accounted for almost 10% of the total population on day 3, when it is unlikely that there is any nutrient or spatial stress[23]. This is also reflected when measuring the remaining essential nutrients of the medium, $NO_3^-$ and $PO_4^{3-}$, which amounted to 96.7% and 83.7% of their original levels, respectively, on day 5 and to 47.3% and 35.6% of the original levels each on day 20 (Supplementary Fig. 1). Since phenotypic diversification occurred shortly after inoculation, and cell density was relatively low, it can be assumed that no spatial stress was present either.

To directly demonstrate that the observed early occurrence of phenotypic diversification is not triggered by external stress and thus differs from the traditional stress response, we examined and compared the performance of mobile populations in the absence and presence of external stress (NaCl). In general, we found distinctly different population dynamics between treatments in terms of population growth and mobility-based population composition (Supplementary Fig. 2). Specifically, compared to normally growing populations in the absence of NaCl, all other populations exposed to NaCl gradients showed different levels of growth arrest and cell death (ANOVA, $F_{4,10} = 213.1$, $p < 0.001$, Supplementary Fig. 2a). In terms of population composition, in contrast to the more stable population composition of populations grown without external stress (approximately 9.4% of the population was non-mobile), relatively low stress (3 g/L NaCl) resulted in a 50:50 ratio of both phenotypes from day 1 onwards, and thus in a significantly higher proportion of non-mobile cells on day 7 (51.7%, $t$ test, $p < 0.01$), while all cells exposed to even higher NaCl concentrations (6, 9, 18 g/L) lost mobility on day 2 (Supplementary Fig. 2b). These results show that the population performance, i.e., growth and population composition, in the absence of external stress differs from population performance under stress. Thus, the traditional stress response paradigm cannot explain the early phenotypic diversification. Instead, we propose that this phenomenon is a bet-hedging behaviour.

In addition, to understand the degree of replication of the phenotypic diversification behaviour (beyond the strain used), we investigated whether the observed phenotypic diversification was strain-specific. To this end, we performed culture experiments under favourable conditions with three additional *H. pluvialis* strains (Supplementary Fig. 3a). We found that all strains tested showed similar patterns of early phenotypic diversification (Supplementary Fig. 3b), suggesting that our results are not strain-specific but a general behaviour of *H. pluvialis*.

Mobile and non-mobile cells differ in both behaviour and morphology. Visible to the naked eye, non-mobile cells lost their mobility and settled at the bottom of the Erlenmeyer flask with a layer of translucent medium on top (enriched non-mobile cells, Fig. 1d, left), which is clearly different from a mobile population with a homogeneous green colour and where no precipitation is observed (enriched mobile cells, Fig. 1d, right). In contrast to the free-living swimming lifestyle of the mobile cells (Fig. 1e, right), we also observed under the microscope that non-mobile cells lost two flagella and most cells aggregated into clumps (Fig. 1e, left). Using transmission electron microscopy (TEM), non-mobile cells were characterised by a significantly thicker cell wall than mobile cells (Supplementary Fig. 4).

Another important trait is cell size, changes in which will have profound effects on cell physiology. We first compared cell size between mobile and non-mobile cells and then recorded the dynamics of cell size in parallel with the growth experiment. Our results showed an overlapping size distribution between the two phenotypes (Fig. 1f). Specifically, mobile cells showed a less variable size distribution, ranging from 15 μm to 25 μm in diameter. The non-mobile population generally showed an increasing trend in cell size (15 μm to 50 μm), with 29.1% of cells larger than 25 μm in diameter and some cells up to 50 μm in diameter.

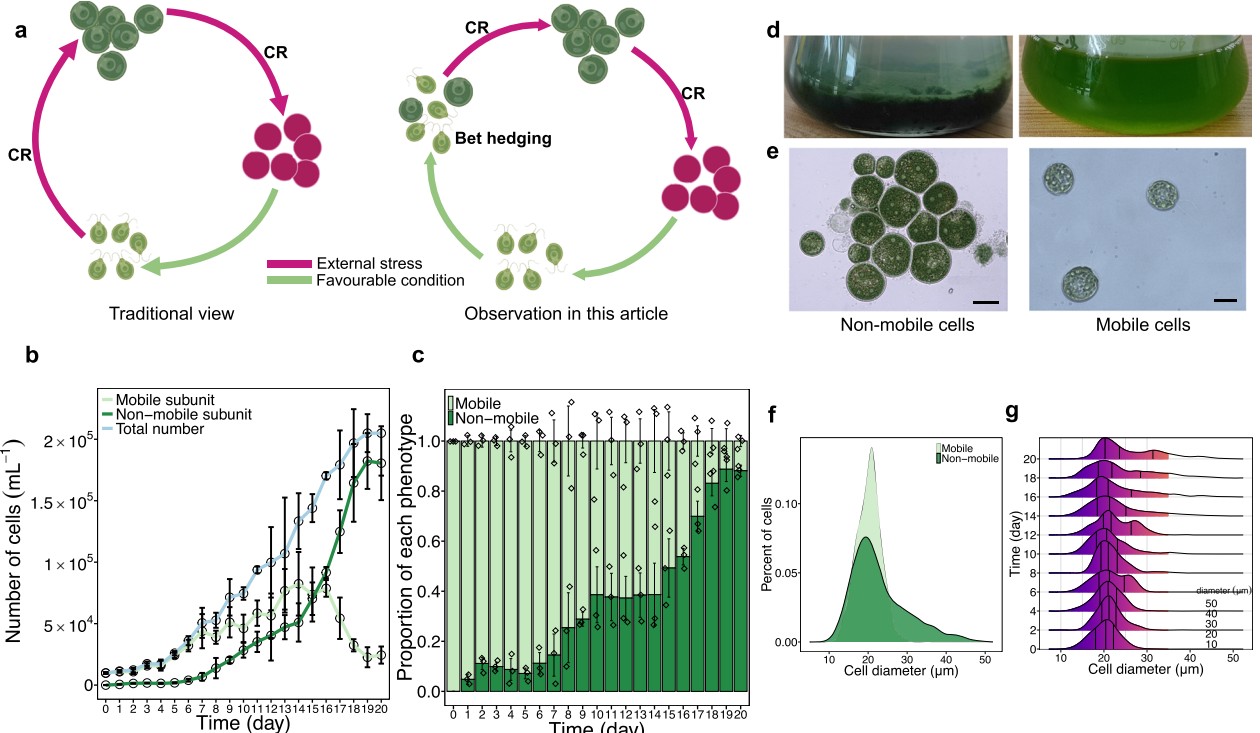

**Fig. 1 | Phenotypic diversification of *H. pluvialis*. a** Graphical summary of the traditional view (left) and the main observation in this study (right). Cells (light green) with two flagella are defined as mobile cells, and bigger round cells (dark green) without flagella are defined as non-mobile cells. Red cells without flagella are defined as non-mobile aplanospores (depicted in dark pink). CR carotenogenic response. **b** Growth curve and phenotypic dynamics over a 20-day period. The blue line shows the total number of cells, the light green and dark green lines show the number of mobile and non-mobile cells, respectively. The graph shows the mean ± standard deviation of the triplicate values. **c** A bar chart showing the change in the ratio of mobile (light green) and non-mobile (dark green) phenotypes over a period of 20 days. The graph shows the mean ± standard deviation of the triplicate values. **d** Images of a culture, artificially enriched for non-mobile cells (left) and a culture, consisting of mobile cells (right). Similar results were observed in four independent experiments. **e** Microscopic images of non-mobile (left) and mobile cells (right) (Scale bar: 20 μm). This experiment was conducted three times independently and similar results were observed. **f** The Gaussian Kernel Density curve illustrating the diameter distribution of mobile (light green) and non-mobile (dark green) cells. **g** The Gaussian Kernel Density curve showing the distribution of diameter on every other day of the growth experiment over a period of 20 days. Source data are provided as a Source Data file.

The cell size dynamics of the growth experiment, with more and more mobile cells converting to non-mobile cells over time, showed a trend towards a gradual increase in the overall population size, with about 25% of the population having a cell size diameter larger than 30 μm on day 20 (Fig. 1g).

## Tradeoff between growth and survival

To test whether the observed heterogeneity in *H. pluvialis* morphology and behaviour is bet hedging, we first investigated whether there is a tradeoff between growth, i.e., increase in cell numbers, and survival, i.e., stress resistance, at the population level. Accordingly, we tested and compared the cell numbers of four populations, i.e., mobile cells only (Mobile), non-mobile cells only (Non-mobile), two artificially created "bet-hedging" populations (20% mobile and 80% non-mobile cells, hereafter M2NM8; 80% mobile and 20% non-mobile cells, hereafter M8NM2), under standard culture conditions, i.e., supernatants from day 10 and under stress conditions (salinity stress, oxidative stress and drought stress) over time. Standard culture conditions leading to lower cell numbers and stress conditions leading to higher cell numbers of the non-mobile population would provide evidence that non-mobile cells trade growth for stress resistance and thus function as the persistent subset of the population.

Under standard conditions, the Mobile population again dissociated into mobile and non-mobile subpopulations, whereas the Non-mobile population maintained the non-mobile state. We observed a 7.9-fold higher cell number (a mixture of mobile and non-mobile cells) in the Mobile population than in the Non-mobile population in an 11-day experiment. The cell numbers of M8NM2 and M2NM8 were lower than those of the Mobile population, but 5.1-fold and 2-fold higher than those of the Non-mobile population, respectively (Fig. 2a). The variance in cell numbers between the two artificially bet-hedging populations manifests the trend that the higher the proportion of mobile cells, the higher the cell numbers.

Although the four populations behaved differently under different environmental stresses, there was a general tendency for the Non-mobile population to have the highest survival rate, the Mobile population to have the lowest survival rate, and the survival rate of the two artificially bet-hedging populations to be in between. When exposed to high salinity (300 mM NaCl) for ten days, the Non-mobile population showed the highest survival rate (48.1%), followed by the two heterophenotypic populations, 22% for M2NM8 and 18.7% for M8NM2, respectively. The Non-mobile population had a 4.9-fold fitness advantage (survival rate) over the Mobile population, which had the lowest survival rate of 9.7% (ANOVA, $F_{3,8} = 156.2$, $p < 0.001$, Fig. 2b). Hydrogen peroxide ($H_2O_2$) treatment was performed to represent oxidative stress, and all populations were more sensitive to oxidative stress than to salinity stress. After $H_2O_2$ exposure (1 mM), survival was highest in the Non-mobile population, at 36%. The M2NM8 population had the second highest survival rate (14.3%). In the M8NM2 and Mobile populations, almost all cells died and the survival rate was 6% and 2%, respectively (Fig. 2c). Drought stress was better tolerated by all populations. Although significantly lower than the survival rate of the other three populations, 42.7% of the Mobile population survived. The Non-mobile population still performed

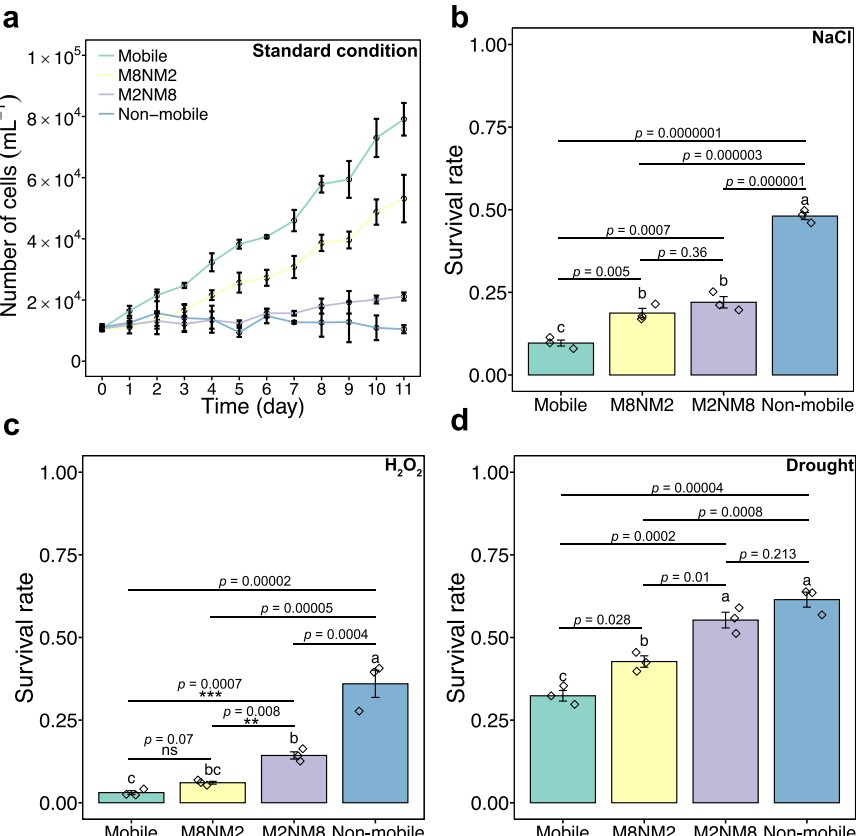

**Fig. 2 | Fitness-relevant parameters of *H. pluvialis* populations with a manipulated phenotypic composition in response to standard and stress conditions.** Growth and survival rates of four populations of different phenotypic composition were evaluated, i.e., Mobile, M8NM2, M2NM8, Non-mobile. **a** Growth of four populations under standard conditions over a period of 11 days. **b** Survival rate of four populations under salinity stress (300 mM NaCl). **c** Survival rate of four populations under oxidative stress (1 mM $H_2O_2$). **d** Survival rate of four populations under drought stress (immediate desiccation). All three survival rate assays under stress conditions lasted for 10 days. The graphs show the mean ± standard deviation of the triplicate values. Different letters on each graph represent statistical significance ($p < 0.05$), calculated by one-way ANOVAs with Tukey's HSD post-hoc analyses across all four populations. Line segments and asterisks in c indicate statistical significance calculated by one-way ANOVA without the Non-mobile treatment. Mobile (mobile cells only, green), M8NM2 (80% mobile and 20% non-mobile cells, yellow), M2NM8 (20% mobile and 80% non-mobile cells, purple), and Non-mobile (non-mobile cells only, blue). Significance: ns (no significance), *($p < 0.05$), **($p < 0.01$), ***($p < 0.001$), ****($p < 0.0001$). Source data is provided as a Source Data file.

best, with a survival rate of 61.4%, followed by 55.3% for M2NM8 and 42.7% for M8NM2 (Fig. 2d).

In summary, these assays provide direct evidence that: (1) the non-mobile phenotype traded growth for stress resistance; (2) the two heterophenotypic populations had lower variance in the two fitness-related traits under different conditions, which were always situated in the middle of two monophenotypic populations. Taken together, this evidence supports our hypothesis of bet hedging.

### Distinct physiological profiles of the two phenotypes

To explore the underlying mechanisms of the higher stress resistance of the non-mobile phenotype, we examined critical physiological parameters in both phenotypes, including photosynthetic activity, cellular dry weight, pigment, lipid and starch content.

Photosynthesis was used as an indicator of stress response, as it is widely accepted that photosynthetic organisms reduce the energy budget in growth-promoting photosynthesis in exchange for investing more energy in activities associated with stress resistance[24,25]. The chlorophyll fluorescence parameter $F_v/F_m$ was tested to determine the maximum quantum efficiency of photosystem II (PSII), an indicator of photosynthetic activity. The result showed that non-mobile cells had a significantly lower $F_v/F_m$ value than mobile cells ($t$ test, $p < 0.05$, Fig. 3a), indicating reduced photosynthetic activity. In addition, the pigments involved in photosynthesis, chlorophyll $a$ and $b$, showed a significantly decreased pattern, accounting for 63.5% and 61.5% of the

value in mobile cells, respectively. Another pigment, carotenoid, was also quantified, and showed an increasing pattern, about twice that of mobile cells (Fig. 3b).

Lipids and starch are considered critical energy storage substances that play a crucial role in response to adverse environmental conditions[26,27]. Higher cellular levels of these substances in non-mobile cells will help us to understand how they achieve higher stress resistance. Cellular lipids, starch, as well as dry weight of non-mobile cells increased significantly by 1.39-fold ($t$ test, $p < 0.05$), 2.46-fold ($t$ test, $p < 0.01$) and 1.95-fold ($t$ test, $p < 0.01$) (Fig. 3c), respectively, compared to mobile cells, indicating an accumulation of these metabolites. Furthermore, an increase in cellular dry weight indicates a shift in energy budget from cell growth to storage. In addition, the ratio of lipids (to dry weight) increased significantly from 20.5% to 36.2% ($t$ test, $p < 0.05$), while the ratio of starch to dry weight rose from 7.2% to 10% ($t$ test, $p < 0.01$) (Fig. 2d), suggesting that lipids are the primary energy storage substance and starch may have an anaplerotic function.

### Distinct transcriptomic profiles between phenotypes

To understand the underlying molecular background and to capture the transcriptomic characteristics of each phenotype, the respective cells were sequenced to examine the key differentially expressed genes (DEGs). In general, 6128 genes of the non-mobile cells showed statistically significant changes in mRNA level compared to the mobile cells (adjusted $P$ value < 0.05), with the transcript level of 2953 genes

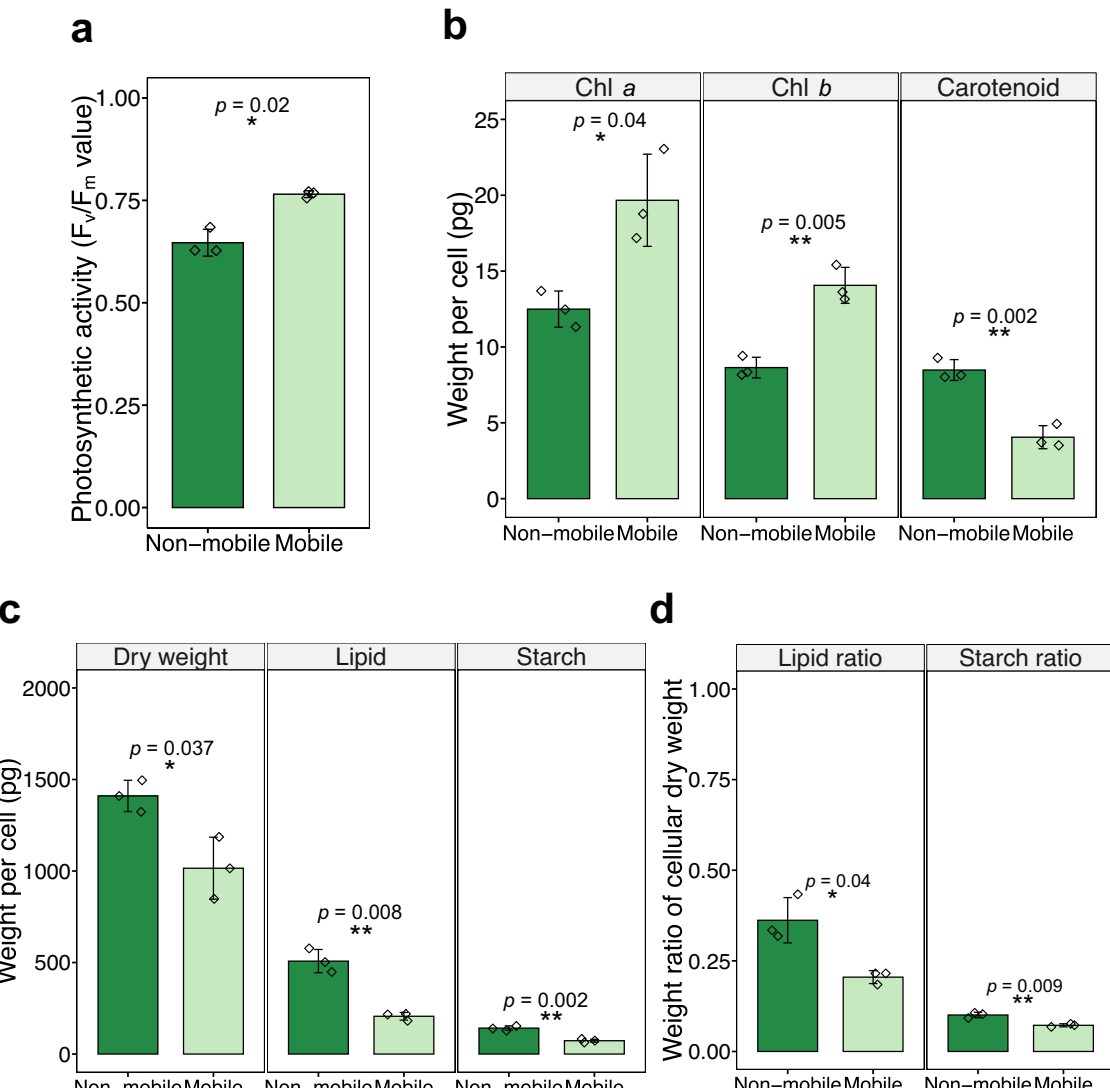

**Fig. 3 | Physiological profiles of mobile and non-mobile *H. pluvialis* phenotypes.** **a** Chlorophyll fluorescence parameter $F_v/F_m$. **b** Cellular chlorophyll *a*, chlorophyll *b* and carotenoids content. **c** Cell dry weight, lipid and starch content. **d** The ratio of cellular lipid and starch to dry weight. The graphs show the mean ± standard deviation of the triplicate values. Asterisks represent statistical significance calculated by two-tailed Welch's *t* test. Mobile cells (light green), non-mobile cells (dark green). Significance: *($p < 0.05$), **($p < 0.01$), ***($p < 0.001$), ****($p < 0.0001$). Source data are provided as a Source Data file.

increased and 3185 genes showing a decreased expression pattern (Fig. 4a). According to the results of the KEGG enrichment analysis, all DEGs were classified into several functional categories, showing striking metabolic differences, such as in photosynthesis, carotenoid biosynthesis, pyruvate metabolism, and fatty acid (a major component of lipids in microalgae) and starch metabolism (Fig. 4b, top 20 pathways). Particular attention was paid to the expression level of specific genes of four major metabolic pathways, i.e., photosynthesis, carotenoid, lipid and starch metabolisms, which are thought to be associated with the cellular stress response. The expression patterns of key genes involved in these four metabolic pathways could clarify at the molecular level how the non-mobile phenotype proves to be more stress-resistant.

In detail, in photosynthesis, genes involved in the synthesis of antenna proteins (a set of important proteins for the assembly of the light-harvesting complex), such as LHCA 1, 3, 4, 5 and LHCB 2, 4, 5[28,29], showed a decreased pattern in non-mobile cells (Fig. 4c), indicating reduced photosynthetic activity. The decreased transcription levels of the antenna proteins are consistent with the $F_v/F_m$ assay, where the non-mobile cells had significantly lowered $F_v/F_m$ levels. As for the

carotenoid metabolism, critical genes involved in carotenoid biosynthesis were upregulated, while genes associated with carotenoid degradation showed a decreasing pattern. For example, the core carotenoid biosynthesis genes *crtB, crtW, lcyB, PDS* and *ZDS*[30,31] showed significantly higher expression levels. Instead, the expression of ZEP (zeaxanthin epoxidase), an important carotenoid regulation gene[31,32] that contributes to the maintenance of normal carotenoid levels, decreased. A similar transcriptomic pattern was observed in the lipid metabolism. Two prominent lipid biosynthesis gene families, *acc* (*accB, accC, accD*) and *fad* (*fadD, fadG, fadH, fadL, fad3, fad4*)[33], were significantly upregulated, while the lipid degradation gene *paaH*[34] was downregulated. The transcription level of related genes suggests that non-mobile cells accumulate carotenoids and lipids. The starch synthase *SS*[35] and the gene *WAXY*[36], associated with starch granule biosynthesis, were markedly upregulated, suggesting accumulation activity. Surprisingly, we also found that several genes involved in starch degradation (*AMY, BAMY, CBM20, treX, malQ*)[35,37–40] significantly increased their transcript levels.

Furthermore, the transcriptomic levels of mitogen-activated protein kinase (MAPK) genes were evaluated to explore whether

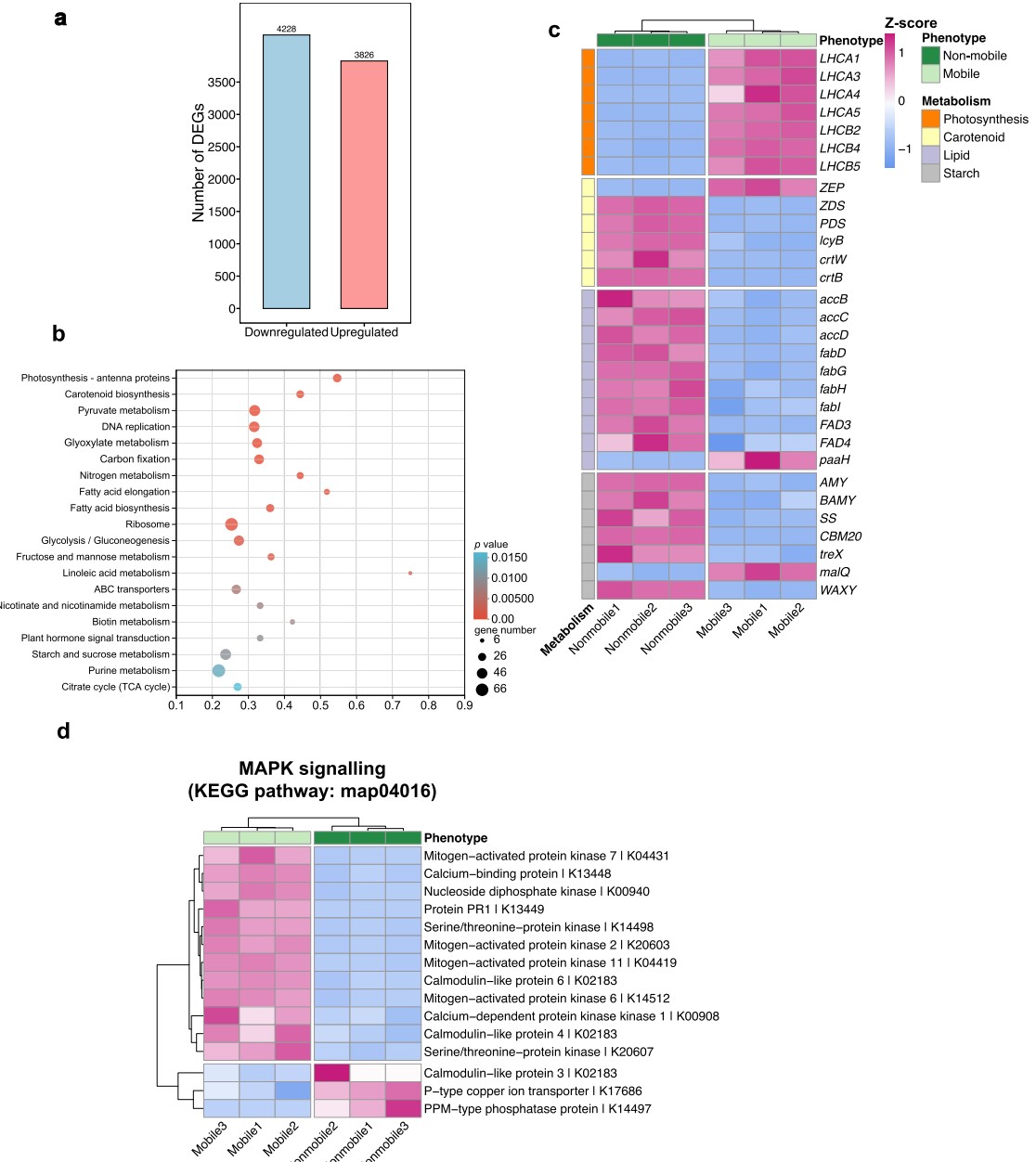

**Fig. 4 | Transcriptome and core functional enrichment analysis of differentially expressed genes in *H. pluvialis*. a** Bar chart of differentially expressed genes of non-mobile cells compared to mobile cells. The numbers on the top of the bar indicate the number of genes that were significantly up- or downregulated. **b** Dot plot analysis of enriched KEGG Pathways with differentially expressed genes with *p* value < 0.05 (right-tailed Fisher's exact *t* test followed by a Benjamin-Hochberg (BH) adjustment). The size of the dot indicates the number of genes for each pathway, and the colour bar indicates the *p* value. **c** Heatmap analysis showing DEGs enriched for four KEGG pathways, including photosynthesis (orange), carotenoid (yellow), lipid (purple), and starch (grey) metabolisms, displayed separately for replicates of the non-mobile (dark green) and mobile (light green) phenotypes. **d** Functional

enrichment analysis of the MAPK signalling pathway displayed separately for replicates of the non-mobile (dark green) and mobile (light green) phenotypes. The heatmap colour gradient is indicative of low gene expression (blue) and high gene expression (dark pink). Heatmap analysis of DESeq2-normalized gene expression scaled as the number of standard deviations from the row means for all genes with KO terms under the KEGG MAPK signalling pathway (map04016). KEGG annotations were assigned from the genome annotation. Column dendrograms show similarity based on Euclidean distance and hierarchical clustering. Gene clusters were determined by k-means clustering with Euclidean distance. Source data are provided as a Source Data file.

non-mobile cells are induced by external stressors. MAPK cascades are evolutionarily conserved signalling modules that serve to convert environmental stimuli into intracellular responses[41]. Various stressors such as temperature, drought, salinity, UV irradiation and reactive oxygen species can activate MAPK signalling pathways[42]. Cellular behaviours regulated by MAPKs are thought to play an essential role in defence responses[42,43]. The expression pattern of the annotated MAPK genes (12 out of 15 annotated genes show a

relatively down-regulated pattern in the non-mobile phenotype, Fig. 4d) illustrates that the transformation of the non-mobile cells was probably not caused by stress to which the mobile cells were exposed, at least not on the day of sampling (day 10). Given that the transformation into the non-mobile state takes time, those transformed ancestor mobile cells at earlier time points should not be stressed either. This result suggests that the formation of non-mobile cells and accompanying physiological modifications may occur

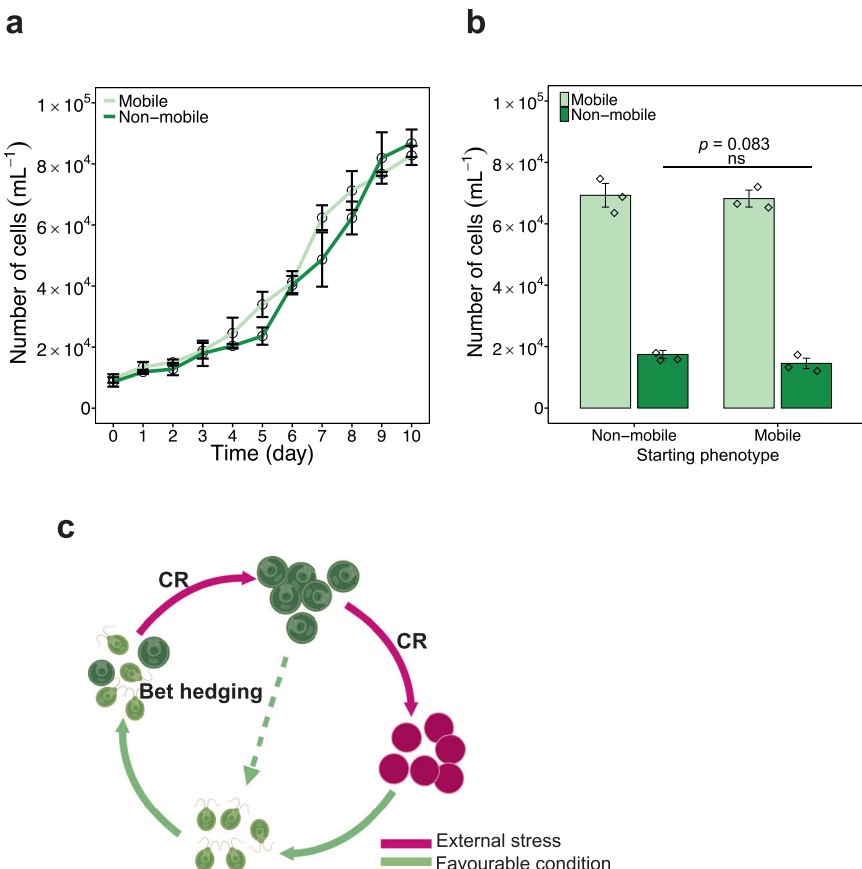

**Fig. 5 | Rediversification of the mobile and non-mobile phenotypes in *H. pluvialis*.** **a** Growth of the population consisting of only mobile (light green) or non-mobile (dark green) cells in fresh medium (BBM) over a period of 10 days. The graph shows the mean ± standard deviation of the triplicate values. **b** Population composition on day 10 for the differently initiated populations. The graph shows the mean ± standard deviation of the triplicate values. Mobile cell population only (light green), non-mobile cell population only (dark green). Statistical significance was calculated using two-tailed Welch's *t* test. Significance: ns (no significance). **c** Graphical representation of the reversibility of the non-mobile phenotype to a mobile state (dashed line). Cells (light green) with two flagella are defined as mobile cells, and bigger round cells (dark green) without flagella are defined as non-mobile cells. Red cells without flagella are defined as non-mobile aplanospores (depicted in dark pink). CR, carotenogenic response. Source data are provided as a Source Data file.

stochastically (stochastic gene expression), another feature of the bet hedging theory[5].

### Ability of both phenotypes to re-diversify

To distinguish between "bet hedging", "evolutionary radiation"[44], "genetic mutation"[45] and "cellular age"[46], all of which can lead to phenotypic diversification, we determined whether mobile and non-mobile cells could re-diversify when re-cultured for another round of growth. The results showed that the mobile population diversified back into a heterophenotypic population. In addition, the non-mobile cells reverted to a mobile status and first regained fertility, then diversified back to the non-mobile phenotype over time (Fig. 5a, b). The phenotype of the founding cells also had no effect on this diversification: both populations initiated with mobile and those initiated with the non-mobile phenotype formed diversified phenotypes, with non-mobile cells constituted averaging 20.1% and 17.6% of the total population (*t* test, *p* > 0.05, Fig. 5b), respectively. These data suggest that the emergence and re-emergence of phenotypic diversification are not due to evolution, mutation or ageing but to bet hedging.

### Discussion

Compared to adaptive tracking and phenotypic plasticity[47,48], bet hedging is an alternative strategy for adapting to randomly fluctuating environments. For a genetically clonal population, it might be advantageous to hedge the bet of population fitness on differentiating into different phenotypes. The presumed advantage of different phenotypes with different survival chances and reproductive success is that they are best suited to different conditions. Consequently, a subpopulation will already exhibit the appropriate phenotype for the new environment before the external environment changes, ensuring that at least a subpopulation can survive and reproduce successfully. Currently, bet hedging is widely reported to be a common adaptive strategy in various organisms. However, to our knowledge, it has never been reported and experimentally verified in microalgae, key players in aquatic ecosystems.

Interestingly, it has been repeatedly reported that the green stage of *H. pluvialis* consists of two phenotypically distinct types at the population level: mobile and non-mobile[17,20,49]. Such phenotypic heterogeneity has been described for a number of other systems[6,12,13,50], and is thought to result mainly from adaptation to microscale environmental variation[5,12,51].

In the case of *H. pluvialis*, it has been suggested that non-mobile cells arise from mobile cells due to external stress[17,18]. This may indeed be the case in older cultures due to increased stress related to high cell density, specifically reduced availability and increased competition for nutrients, self-shading, and the release and accumulation of unfavourable substances from dead cells[52,53]. Under these circumstances, it is reasonable to expect cells to switch from growth-dedicated activities to stress-resistant metabolisms (the non-mobile phenotype). In our study, we observed an increasing proportion of non-mobile cells over time (Fig. 1b, c), and in particular an acceleration of the transformation from mobile to non-mobile cells from day 8 onwards, becoming very

apparent from day 14 onwards (Fig. 1c). At this relatively late stage of growth, we agree that external stress occurred and led to the observed shift in the population, which was then soon dominated by the stress-resistant non-mobile state.

However, this traditional stress sensing and response does not explain the early appearance of non-mobile cells at the beginning of the culture period and cannot explain why only a small fraction of the population changes, while the majority remains mobile (Fig. 1b, c), suggesting that this phenotypic diversification is not triggered by stress, at least at these early time points. Indeed, when mobile populations were stressed by culturing at different concentrations of NaCl, they showed distinct growth and population compositions compared to the non-NaCl populations (Supplementary Fig. 2). This demonstrates that the observed early phenotypic diversification was not induced by stress, which is further supported by the relatively down-regulated MAPK pathway genes (stress-dependent signalling pathway) of the non-mobile cells (Fig. 4d), suggesting that the non-mobile cells do not suffer from stress and may not have arisen as a response to stress.

The observed phenotypic heterogeneity of *H. pluvialis* can be explained by bet hedging, where mobile cells represent the fast-growing subunit and non-mobile cells represent the stress-resistant subunit of the population. At the expense of reduced population growth caused by the transformation of some cells to a slow/non-growing, non-mobile phenotype (given the high proportion of non-mobile cells less than 20 μm in diameter at day 20 (Fig. 1f), further evidence is needed to clarify whether non-mobile cells can reproduce). Bet-hedging populations maintain a balance between population growth and stress resistance, ensuring that both population reproduction and population survival are non-zero. Notably, the bet-hedging behaviour described here differs from the traditional stress response, i.e., the well-known carotenogenic response of *H. pluvialis*[21]. Most importantly, the non-mobile green cells are always present, even in the early stages of the cultures, when there is no stress and the majority of the culture is in the mobile state.

Apart from the marked difference in mobility, the two phenotypes also differ in other traits, such as cell size and grouping behaviour. Non-mobile cells generally have a larger cell diameter and, in contrast to free-living mobile cells, most of them aggregate into clumps. Such aggregation behaviour could serve to create a stable microenvironment or to defend against size-dependent predators. In addition, we observed a significantly thicker cell wall in non-mobile cells compared to the cell wall of mobile cells (Supplementary Fig. 4), which likely acts as a physical barrier protecting the cells from external harmful chemical exposure or physical damage.

We also evaluated population growth and survival as indicators of fitness in four artificially created populations that differed in their ratio of mobile to non-mobile types under standard and stress conditions. The results were consistent with our expectations, as: (1) the non-mobile phenotype traded growth for stress resistance, and (2) both bet-hedging populations showed lower variance in the fitness-related parameters across environments (Fig. 2). Furthermore, the ability of each phenotype to re-diversify suggests that the phenotypic heterogeneity in this study is due to bet hedging rather than evolutionary radiation, mutation or ageing; a similar phenomenon has been observed in the bet-hedging bacterium *Sinorhizobium meliloti*[13]. Taken together, our results provide strong evidence that the phenotypic heterogeneity of *H. pluvialis* serves as a bet-hedging strategy.

It is of general interest to understand how the non-mobile subpopulation becomes more stress-resistant. In the study of microalgae for stress resistance, it is well documented that persistent cells typically reduce photosynthetic output and other non-essential metabolic activities and instead increase metabolic processes associated with persistence. For example, the accumulation of antioxidants and energy storage materials, including lipids and starch, is thought to be critical

for resistance to environmental stress[27,31,54,55]. We therefore investigated the physiological and transcriptomic profiles of the non-mobile phenotype, focusing on these metabolites, to uncover the underlying mechanisms of stress resistance. In general, the integrated physiological and transcriptomic data show that non-mobile cells have shifted from growth-promoting photosynthesis to resistance-promoting activities, including the accumulation of antioxidant and energy-storing substances.

In detail, our results show that non-mobile cells significantly decreased photosynthetic activity, based on the $F_v/F_m$ assay (Fig. 3a), which might result from the decreased amount of chlorophyll *a* and *b* (Fig. 3b) and the decreased expression of light-harvesting complex proteins (Fig. 4c). Instead, cellular carotenoids (antioxidative substances)[56], lipids and starch (energy storage compounds)[26], all increased significantly, as expected (Fig. 3b, c). This is confirmed by the transcriptomic analyses and suggests a tradeoff between growth and stress resistance. Counterintuitively, five genes related to starch degradation (*AMY, BAMY, CBM20, treX, malQ*) also showed higher expression in non-mobile cells (Fig. 4c), making the results intriguing to interpret. We propose that the accumulation of starch could serve as a rapid response to stress and that starch acts as an intermediate energy storage compound, which could later be used to provide energy for other subsequent processes, such as lipid biosynthesis, which is considered a long-term energy storage. This unclear gene expression pattern could explain why lipids accounted for 36.2% but starch only for 10% of the cell dry weight (Fig. 3c). Taken together with the near arrest of growth, increased cell size, aggregation behaviour and thickened cell wall (Fig. 1e, f, Supplementary Fig. 4b), all traits provide evidence for why non-mobile cells are more stress resistant than their mobile counterparts.

An almost ubiquitous limitation of bet hedging studies is the understanding of the underlying mechanisms of bet hedging[57], which also proves to be challenging in *H. pluvialis*. To date, the most popular explanation for the occurrence of bet hedging is stochastic initiation of gene transcription[6,13]. In our case, the early diversification of phenotypes is a simple but important indicator of stochastic gene expression (detailed by the transcriptomic analyses, Fig. 4), but it would also be very interesting to explore the evolutionary and molecular origins of this phenomenon.

More interestingly, in contrast to other reports of microbial bet hedging, where only one stress-resistant phenotype exists, in *H. pluvialis*, the final resistant form is the red aplanospore (carotenogenic response)[17,18,58]. Non-mobile green cells continue to develop into aplanospores as the environment continues to deteriorate. Red aplanospores are thought to be able to survive extremely harsh environmental conditions, although here we show that the non-mobile cells are already stress resistant. The existence of two different resistance phenotypes would make sense from an evolutionary point of view, as the non-mobile green cells represent a rapid response to the environment and can quickly revert to a fast-growing state, i.e., the mobile phenotype, when conditions improve (Fig. 5). If environmental conditions continue to deteriorate, the non-mobile green cells transform into aplanospores, the ultimate resistant form that can survive harsher conditions for longer.

The presence of two progressive persistence phenotypes could serve as an essential and effective adaptation strategy for *H. pluvialis* in nature, which would have been selected by the species' complex natural habitats. Indeed, unlike microalgal species living in open waters, the natural habitats of *H. pluvialis* are often small, temporary natural or artificial aquatic habitats[59,60], whose conditions can easily change and dramatically alter, posing extreme challenges to the organisms living in them. Under such conditions, the survival and reproduction of the *H. pluvialis* population, and the subsequent competition for nutrients and space, would particularly favour bet hedging. By forming mobile and non-mobile subgroups, the *H. pluvialis* population could take

advantage of changing environmental conditions that can be devastating to other microalgal species, while increasing their chances of survival until the environment becomes favourable again.

Similar to the challenges in identifying bet-hedging behaviour in bacteria, microalgal bet hedging is relatively difficult to detect because it can be hard to identify phenotypic diversification or metabolic variation and then isolate the corresponding phenotypes of a population. This is likely to be the reason why, although it has been occasionally suggested[61], no experimental studies have yet provided solid experimental evidence for bet hedging. While we do not claim to demonstrate the universality of bet hedging in microalgae, our results show a general bet-hedging behaviour of *H. pluvialis*, as supported by the fact that all four *H. pluvialis* strains showed similar patterns of early phenotypic diversification (Supplementary Fig. 3), and it is very likely that bet hedging behaviour of microalgae is more widespread and frequent than expected. For example, some blooming microalgal species that form dormant cysts as part of their life history can not only survive global environmental changes, but also thrive even in the face of human interventions such as physical, chemical and biological anti-blooming attempts. To some extent, this may be due to the unrecognised bet hedging of these species, i.e., the simultaneous coexistence of vegetative cells and cysts, which allows them to survive anthropogenic disturbance and recover again.

In summary, here, we provide evidence that the observed phenotypic heterogeneity in *H. pluvialis* is a bet-hedging strategy in which parts of the mobile population spontaneously and reversibly transform into a distinct, non-mobile state that exhibits almost no growth but significantly higher resistance to stress. Measurements of fitness proxies, i.e., population growth and survival rates, show that hetero-phenotypic populations have reduced variance in these parameters under different conditions, consistent with the basic concepts of bet hedging. Further analyses indicate that the ability of the non-mobile phenotype to withstand stress is due to metabolic shifts from growth-promoting photosynthesis to stress-resistant activities such as cell enlargement and aggregation, thickened cell wall, and the accumulation of antioxidative substances (carotenoids) and energy storage compounds (lipid and starch). Overall, this study provides an experimental account of bet-hedging behaviour in the phyla of microalgae, providing fundamental insights into how these organisms can cope with unpredictable, fluctuating environments, offering a different perspective for future species or ecosystem rescue or management, and prompting us to rethink the mass cultivation strategy of this commercially valuable species.

## Methods

### Strain and culture condition
*H. pluvialis* (FACHB-712, Freshwater Algae Culture Collection at the Institute of Hydrobiology, Wuhan, China), also known as *Haematococcus lacustris*, was maintained photoautotrophically in freshwater BBM medium as previously described[62]. Specifically, the stock culture was maintained in 500 mL of freshwater medium BBM in Erlenmeyer flasks at $25 \pm 1\,°C$ under 12 h/12 h light–dark cycles (20 μmol photons $m^{-2}\,s^{-1}$). The cells were grown without forced aeration, the only $CO_2$ supply being diffusion from the atmosphere into the flasks.

### Growth experiment, phenotype and cell size dynamics
Prior to the growth experiment, the stock culture was first transferred to fresh BBM, and the mobility of the population was checked daily under an inverted microscope (Primo Vert, Carl Zeiss, Germany). From experience, a high proportion of cells (>95%) show the ability to swim in the first few days when transferred to fresh medium, even if it was initiated with a heterophenotypic population composed of mobile and non-mobile mother cells. Because non-mobile cells lose their swimming ability and will sediment to the culture flask bottom, by gently pipetting the upper-level culture after standing still for 1 h, we can

enrich the mobile cells. In this way, a 100% mobile population was achieved, which was used in the subsequent growth experiment.

In the growth experiment, an initial density of about $1*10^4$ cells/mL of cells was inoculated into 200 mL of fresh medium in an Erlenmeyer flask at $25 \pm 1\,°C$ under 12 h/12 h light–dark cycles (20 μmol photons $m^{-2}\,s^{-1}$) for 20 days in triplicate. Cell density and population composition based on mobility were recorded daily. Specifically, 100 μL culture was sampled after careful mixing and counted with a counting chamber under an inverted microscope. Mobile cells and non-mobile cells were distinguished from each other and recorded on the basis of observing the mobility of an individual after 5 min of accommodation in the counting chamber to exclude the effects of physical disturbance during sampling.

Meanwhile, another 100 μL culture was used to measure cell size. A thin layer of low melting point agarose gel (Thermo Fisher Scientific, USA) was prepared to stabilise the cells. Images were then captured and processed using AxioVison SE64 Rel.4.9 software, where cell diameters were determined using the length measurement function. For each replicate at each time point, cell diameters of approximately 300–500 cells were measured.

### Separation of subpopulations and determination of cell size
Due to the precipitating and clustering properties of the non-mobile phenotype, the two cell types are relatively easy to isolate and enrich. For the isolation of the mobile cells, 10-day-old populations of the growth experiment were kept still for 1 h to allow the non-mobile cells to settle to the bottom, and the upper green part of the culture, containing the mobile cells was pipetted out and enriched into new flasks. For the non-mobile cell isolation, the remaining mobile cells were resuspended in 20 mL of added double distilled water (ddH$_2$O, ipureplus, neoLab Migge, Germany), after which the liquid was pipetted out and discarded. Another 20 mL of ddH$_2$O was used to wash off the remaining non-mobile cells by pipetting. The separate mobile and non-mobile cultures were checked under the microscope to ensure that the two phenotypes did not coexist. To determine the cell size of each phenotype, the cell diameter of the two separate cultures was examined under the microscope as described above.

### Nutrient dynamics over time
Residual inorganic nutrients (nitrate and phosphate) of the culture medium were measured on day 0, day 5, day 10 and day 20 using the Auto Discrete Analyzer (Cleverchem Anna, Dechem-Tech, Germany). In particular, nitrate and phosphate concentrations were measured using the cadmium-zinc reduction method and molybdenum-antimony spectrophotometry, respectively[63].

### Phenotypic dynamics of different *H. pluvialis* strains
An initial density of about $1*10^4$ cells/mL of mobile cells of four *H. pluvialis* strains, i.e., FACHB-1928, FACHB-712, FACHB-827, FACHB-871 purchased from the Freshwater Algae Culture Collection at the Institute of Hydrobiology, Wuhan, China, was inoculated into 200 mL of fresh BBM medium at $25 \pm 1\,°C$ under 12 h/12 h light-dark cycles (20 μmol photons $m^{-2}\,s^{-1}$) for 14 days in triplicate. Population composition based on mobility was recorded daily as described above.

### Population performance under NaCl stress
To distinguish conventional stress response from the proposed bet-hedging behaviour of *H. pluvialis* in this study, an initial density of about $1*10^4$ cells/mL of mobile cells was inoculated into 200 mL of fresh medium with the addition of 0, 3, 6, 9 and 18 g/L NaCl, respectively. Cultures were grown at $25 \pm 1\,°C$ under 12 h/12 h light-dark cycles (20 μmol photons $m^{-2}\,s^{-1}$). Cell density and population composition based on mobility were recorded daily. The experiment lasted for 7 days and was done in triplicate. Cell density and population composition based on mobility were recorded daily.

## Microscopy

Microscopic images of mobile and non-mobile cells were taken using an inverted fluorescence microscope (Axio observer Z1, Carl Zeiss, Germany).

For transmission electron microscopy (TEM), respective mobile and non-mobile cells were first fixed by 2.5% glutaraldehyde for 4 h at 4 °C. These cells were then rinsed three times with PBS (pH 7.4) and were post-fixed with 1% $OsO_4$ for 2 h at 20 °C, followed by another round of rinsing three times with PBS (pH 7.4). Samples were then dehydrated and infiltrated with epoxy resin, followed by polymerisation at 60 °C for 48 h. Ultrathin sections were cut on a Leica UC7 (Leica, Vienna, Austria) using a diamond knife (Daitome Ultra 45°, USA). The sections were mounted on copper grids, stained in saturated 2% uranium acetate (15 min) and dried at 20 °C overnight. Sections were observed and images were captured at 100 kV with a transmission electron microscope (JEM-1400, JEOL, Japan).

## Growth and survival assays under various conditions

Four populations were included in the growth and survival assays, i.e., mobile cells only population (Mobile), non-mobile cells only population (Non-mobile), as well as two artificially prepared bet-hedging populations (20% mobile and 80% non-mobile cells, hereafter M2NM8; 80% mobile and 20% non-mobile cells, hereafter M8NM2).

To compare growth and survival of each population under standard culture conditions, supernatants of cultures of the growth experiment (day 10) were first collected by centrifugation at $2348 \times g$ (Centrifuge 5424R, Eppendorf, Germany) for 2 min followed by syringe filtration (0.22 μm). Then, a starting density (-1*10^4 cells/mL) of each population was transferred into 100 mL of supernatant. Cell number was checked with a cell counting chamber under the microscope every 24 h. The experiment lasted for 11 days and was done in triplicate.

For growth and survival assays under stressful conditions, salinity stress (extra 300 mM sodium chloride, NaCl, Karl Roth, Germany), oxidative stress (1 mM hydrogen peroxide, $H_2O_2$, Karl Roth, Germany) and drought stress were included. For salinity stress and oxidative stress, a starting density of -1*10^4 cells/mL of each population was cultured in 200 mL of customed supernatant, respectively. For drought stress, 2 mL culture (-1*10^4 cells/mL) of each population was spread on a 0.22 μm membrane filter (round shape, 47 mm in diameter, Thermo Fisher Scientific, USA) upon a kitchen paper to remove the liquid. The filter membrane was then transferred into a petri dish and dried in the air. All triplicated treatments were exposed to stresses for 10 days, after which the surviving cells were enumerated under the microscope. Viable cells were discriminated from dead cells based on their intact cell shape and autofluorescence under the microscope. Unless otherwise specified, other culture parameters were $25 \pm 1$ °C under 12 h12 h light–dark cycles (20 μmol photons $m^{-2}$ $s^{-1}$).

## Quantification of cell dry weight, chlorophyll, carotenoids, starch and lipids

For the determination of cell dry weight, 10 mL of culture was collected in a pre-dried centrifuge tube and centrifuged at $3381 \times g$ for 5 min (Centrifuge 5424R, Eppendorf, Germany). The cell pellets were washed three times with deionised water, dried for 3 h at 105 °C in an oven (Yiheng, Shanghai, China) and then cooled to room temperature before weighing.

Chlorophyll $a$, chlorophyll $b$ and carotenoid contents were measured according to a previously published method[64–66]. In brief, 10 mL of the culture of each phenotype was centrifuged at $18,407 \times g$ for 10 min at 4 °C (Centrifuge 5424R, Eppendorf, Germany). The supernatant was discarded, and the remaining cells were resuspended in 2 mL of 90% methanol (Aladdin, Shanghai, China). The suspension was incubated in a water bath at 70 °C for 10 min and centrifuged again at $18,407 \times g$ (Centrifuge 5424R, Eppendorf, Germany). The supernatant was used to measure absorbance measurement with an ultraviolet-visible (UV–Vis) spectrophotometer (UV-1780, SHIMADZU, Japan) at wavelengths of 470, 652 and 665 nm. The pigment content was determined using the following formulas:

$$Chlorophyll\ a\ (mgL^{-1}) = 16.82A_{665} - 9.28A_{652};\qquad(1)$$

$$Chlorophyll\ b\ (mgL^{-1}) = 36.92A_{652} - 16.54A_{665};\qquad(2)$$

$$Carotenoid\ (mgL^{-1}) = (1000A_{470} - 1.19C_a - 95.15C_b)/225;\qquad(3)$$

where $A_{470}$, $A_{652}$ and $A_{652}$ are the absorbances at wavelengths of 470, 652 and 665 nm, respectively.

Cellular starch was quantified according to a published method with some modifications[67]. Three millilitres of each sample was ground with a grinding rod. The pigments were extracted three times with 4 mL of 80% ethanol for 15 min at 70 °C. For complete hydrolysis of the starch, 0.6 mL of 52% perchloric acid was added to the precipitate, stirred for 30 min at 25 °C and centrifuged. This procedure was repeated three times. Then 0.2 mL of the extract was cooled to 0 °C; 1 mL of anthrone solution [0.1 g of anthrone in 50 mL of 82% (v/v) $H_2SO_4$] was added and stirred. The mixture was kept in a water bath at 100 °C for 8 min and then cooled to 20 °C. The absorbance was measured at 625 nm. Calibration was performed simultaneously with glucose as the standard.

The intracellular lipid content was determined with a neutral lipid-specific dye, Nile Red (9-diethylamino-5H-benzo(a)phenoxazine-5-one)[68]. Specifically, lipid content was determined by breaking cells with a grinding rod followed by extraction with chloroform/methanol (2:1, v/v) and total lipid content was measured gravimetrically. As for the samples, 40 μL of Nile Red solution in acetone (250 mg/L) was added to 1 ml of algae suspension (20% DMSO). The mixture was vigorously agitated with a vortex mixer. Fluorimetric analysis was performed 10 min after staining using an Enzyme labelling apparatus (Tecan, Swiss) with a 490 nm narrowband excitation filter and a 585 nm narrow band emission filter.

## Photosynthetic activity

Quantification of photosynthetic performance based on chlorophyll fluorescence was analysed using the FL3500 fluorometer (PSI, Czech Republic). 2 mL of separated mobile and non-mobile cultures were placed in darkness for 20 min, and the maximum quantum efficiency of PSII photochemistry ($F_v/F_m$) was measured immediately after dark adaptation using a PPFD of 3000 μmol $m^{-2}$ $s^{-1}$ as a saturating flash for 1 s duration.

## Transcriptomics

One hundred microliters of each phenotype was harvested as described above and centrifuged for 10 min at $1503 \times g$ at 4 °C (Centrifuge 5424R, Eppendorf, Germany) to remove the supernatant. One hundred microliters of $ddH_2O$ was added to resuspend the cell pellets and centrifuged again; the supernatants were discarded. This procedure was performed twice. The cell pellets were transferred to RNase-free cryogenic vials and frozen in liquid nitrogen for 30 min until extraction. In total, both cell types were analysed in triplicate.

Total RNA was extracted with Plant RNA Purification Reagent for plant tissue according to the manufacturer's instructions (Invitrogen, USA), and genomic DNA was removed using DNase I (TaKara, Dalian). Total RNA samples were sent to Shanghai Majorbio Bio-pharm Technology Co., Ltd (Shanghai, China). RNA quality was determined using a 2100 Bioanalyzer (Agilent) and quantified using the ND-2000 (Nano-Drop Technologies). Only high-quality RNA samples ($OD_{260/280} = 1.8$-2.2; $OD_{260/230} \geq 2.0$; RIN $\geq 6.5$; 28 S:18 S $\geq 1.0$; and $> 1 μg$) were used for sequencing library construction.

The RNA-seq transcriptome library was prepared from 1 µg of total RNA (messenger RNA was isolated according to the polyA selection method by oligo(dT) beads. Thus, gDNA will not interfere with the results) using the TruSeq™ RNA sample preparation kit from Illumina (San Diego, CA, USA). Data were analysed using the Majorbio Cloud online platform (www.majorbio.com). Raw paired-end reads were trimmed and quality controlled by SeqPrep (https://github.com/jstjohn/SeqPrep) and Sickle (https://github.com/najoshi/sickle) using the default parameters. The clean reads were then separately aligned to the reference genome with orientation mode using HISAT2 software (http://ccb.jhu.edu/software/hisat2/index.shtml). The mapped reads from each sample were assembled using StringTie (https://ccb.jhu.edu/software/stringtie/index.shtml?%20t=example) in a reference-based approach.

To identify the differentially expressed genes (DEGs) between two different treatments, the expression level of each transcript was calculated using the transcripts per million reads method (TPM). RSEM (http://deweylab.biostat.wisc.edu/rsem/) was used to quantify gene abundances. Essentially, differential expression analysis was performed using DESeq2 v1.32.0 in R v4.1.1 with a $Q$ value $\leq 0.05$, and DEGs with $|log2FC| > 1$ and a $Q$ value $\leq 0.05$ were considered significant DEGs. In addition, functional enrichment analyses, including Kyoto Encyclopedia of Genes and Genomes (KEGG, https://www.kegg.jp/) analyses, were performed to determine which DEGs were significantly enriched in KO terms and metabolic pathways at Benjamin-Hochberg corrected $P$-value $\leq 0.05$ compared to the whole transcriptome background. Analyses of functional GO enrichment and KEGG pathways were performed by Goatools (https://github.com/tanghaibao/Goatools) and KOBAS (http://kobas.cbi.pku.edu.cn/home.do), respectively.

### Re-diversification of the individual phenotypes
Triplicated mobile and non-mobile populations were each transferred for 10 days into 200 mL of fresh BBM medium at an initial density of $1*10^4$ cells/mL. The cell number and composition of the mobile population were checked daily.

### Statistical analyses
All statistical analyses were performed in R v4.1.1. Statistical significance for physiological measurements and nutrient dynamics was calculated by Welch's $t$ test for pairwise comparisons of two treatments ($p$ value $< 0.05$). A one-way ANOVA with Tukey's HSD post-hoc analysis ($p$ value $< 0.05$) was conducted for all tested populations of growth and survival assays.

### Reporting summary
Further information on research design is available in the Nature Portfolio Reporting Summary linked to this article.

## Data availability
The data of this study are available within the article. Raw RNAseq reads for differential gene expression analyses have been submitted to NCBI's SRA database (http://www.ncbi.nlm.nih.gov) under BioProject PRJNA940855. The KEGG database (https://www.kegg.jp/) was used for functional enrichment analyses. Source data are provided with this paper.

## Code availability
All scripts for data visualisation are available at https://github.com/SiTANG1990/Microalgal-bet-hedging/tree/v1.0.0 (https://doi.org/10.5281/zenodo.10578478[69]).

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

## Acknowledgements

This work was supported by the National Natural Science Foundation of China (41976126) and the S&T Projects of Shenzhen Science and Technology Innovation Committee (JCYJ20200109142822787,

KCXFZ20230731093402005, RCJC20200714114433069, and JCYJ20200109142818589), the Project of Shenzhen Municipal Bureau of Planning and Natural Resources (Grant No. [2021]735-927), as well as the Shenzhen-Hong Kong-Macao Joint S&T Project (SGDX20220530111204028). K.H. gratefully acknowledges the financial support of the John Templeton Foundation (#62220). The opinions expressed in this paper are those of the authors and not those of the John Templeton Foundation.

## Author contributions

S.T., K.H., J.Z. (Jin Zhou) and ZH.C. conceptualised and designed the experiments. S.T., YQ.L., JM.Z. (Jianming Zhu), XY.C. and L.L. performed the experiments. S.T., YQ.L., J.Z. (Jin Zhou) and K.H. analysed and visualised the data. All authors interpreted the results and wrote the manuscript.

## Competing interests

The authors declare no competing interests.
