## [Peer Review File · Nature Communications]

Bet hedging in a unicellular microalgaReviewers' Comments:

Reviewer #1:

Remarks to the Author:

This is a good example of very interesting study shifting the paradigm in the biology of carotenogenic microalgae and the origin of their stress resilience. This study has the potential to be published since it is of certain interest for the audience of the journal, but the manuscript needs some overhauling and re-arrangement of emphases throughout, although the writing is of good quality and the results are technically sound.

The title of the paper is of clickbait nature, but it gives a vague idea of what it is about (see also my comments below).

L20: ...adapt to unpredictable future... sounds a bit like an oxymoron (in a sense it is not really possible for a biological entity to adapt to a thing which did not yet happen). It is possible to prepare for this, to make some arrangements beforehand, but this is not an outcome of a factor which acts at present, it is more on the side of intelligent behavior which is not in the scope of this paper. On the other hand, it is quite possible to adapt to a volatile environment "assuming" that the environment will retain this characteristic in future... It is actually selection for the breadth of reaction norm, in terms of ecology and evolution. Please consider and possibly rephrase: "Secondary carotenogenesis as a bet hedging strategy ... in microalgae..."??? Please see also comments and questions below.

L33: storing -> storing

L74: it is not a strain, it is a species.

Judging from Fig. 1, the increase in immotile cells takes place at the onset of stationary phase of growth (roughly, from the middle of the shown observation period, d 14). Before that, the proportion of different cell types change more or less synchronously. It is accepted that the preparation to the induction of carotenogenic response (including the formation of immotile palmelloid cells) starts when the cell division rate slows down, so the metabolic sink for the photosynthates shrinks. Effectively this situation (displacing the balance of the production of photosynthates and the metabolic demand) resembles the onset of a stress provoking the above-mentioned changes (loss of cell motility and, later on, haematocyst formation). Fig. 1D shows that the cell suspension possessed, at advanced stages of cultivation, a high density, so at this stage the growth was likely limited by light. This is corroborated by the presence of linear phase at the growth curve (Fig. 1B). So the loss of motility can be, at least in part, due to a different stress (low light). Therefore, "bet hedging" may be another interpretation of these well-known changes in the population of carotenogenic microalgae.

L122: what is "spatial stress" for microalgae?

Considering the results summarized in Fig. 2: those provide a solid confirmation of well-know stress resilience of the immotile cells of the carotenogenic microalgae. By the way, did the cell populations further dissociate into motile and immotile cells under the "standard" conditions (Fig. 2A)?

The bottom line is that this study shows the applicability of the theory of "bet hedging" to the populations of microalgal using the illustrious example of carotenogenic microalga *Haematococcus pluvialis*. The most conclusive results in this context are constituted by the observation of the population structure under favorable growth conditions and the experiments on rediversification of the cell types upon re-inoculation into the fresh medium. The other part of the study represents a recapitulation of what is already known on the stress physiology and biochemistry of *H. pluvialis* (see e.g. Boussiba 2000 and Recht 2014 etc.). It would much more to the point to use continuous culture under favorable conditions to show the persistence of the population (re-)differentiation, the limitations inevitable in the batch cultivation experiments complicate reaching this goal. Ultimately, a kind of bet hedging of this sort can be found in many organisms, it is not quite correct to say that it is something newly discovered and only in these microalgae. This phenomenon actually has an established name---secondary carotenogenesis. Therefore, I cannot agree with the statements in LL65-68. It is also recommendable to explore some other non-carotenogenic microalgae species to find out of the so-called bet hedging is associated with the induction of the carotenogenic response (and is actually an alternative designation of it) or it is a more universal phenomenon in microalgae.

Reviewer #2:

Remarks to the Author:

As it has been discussed, phenotypic heterogeneity in green stage of *Haematococcus pluvialis* has been reported in previous studies, although an early appearance was observed in this study. The early presence of non-motile cells was explained by the authors as a bet hedging strategy to deal with unpredictable changes in environment. Variations in growth, responses to adverse environment, physiological profiles as well as transcriptome, were examined between populations with distinct compositions. Although a great deal of interesting data has been provided in the manuscript, they are largely descriptive. No sufficient mechanistic insights into bet hedging in microalgae were disclosed. Additionally, is it possible the early appearance of the phenotypic heterogeneity in this study is a strain-specific phenomena? Conclusively, the work does not provide the level of advancement that the audience would expect, from my point of view. Therefore, I cannot recommend publication of the manuscript.

Reviewer #3:

Remarks to the Author:

The manuscript submitted by Tang et al. presents an elegant ecophysiological study of a unicellular microalga that exhibits a bet hedging. The bet hedging in microorganisms is a fascinating survival strategy that secures populations' existence in fluctuating environments, and this is the first communication of such phenomenon in microalgae. *Haematococcus pluvialis* aside from being an important freshwater primary producer, too has important biotechnological applications. The findings presented by the authors are noteworthy and of interest to a wide audience, even though the mechanism behind the bet hedging phenomenon in *H. pluvialis* remains elusive.

I enjoyed reading the manuscript, it is well written. The data set presented is clear and neat, with sufficient replications and statistical analysis. The methodology used is sound. The work supports conclusions and stated hypotheses.

I have no major comments and I support the publication of the manuscript.

Minor comments:

- although names above the genus level are not always italicized, I encourage italics for names of all taxa regardless of rank,
- in methodology, I suggest using rcf instead of rpm, or provide details such as model of rotor and centrifuge,
- in transcriptomic analysis, the removal of genomic DNA is a crucial, yet tricky step, as a single DNase treatment is not always sufficient and requires checks for residual genomic DNA and eventual repeat. Has the RNA been tested for presence of residual gDNA?
- please check and correct the reference list. Several papers are listed at least twice in the reference list (e.g. Baumont et al., Lycus et al., Philippi et al.)

with regards,

Pawel Lycus

Responses to Reviewer #1

This is a good example of very interesting study shifting the paradigm in the biology of carotenogenic microalgae and the origin of their stress resilience. This study has the potential to be published since it is of certain interest for the audience of the journal, but the manuscript needs some overhauling and re-arrangement of emphases throughout, although the writing is of good quality and the results are technically sound.

Thank you for the detailed response and your thoughtful comments! We have implemented your comments in the manuscript throughout. Please see our detailed responses below.

The title of the paper is of clickbait nature, but it gives a vague idea of what is it about (see also my comments below).

We respectfully disagree. Based on additional experiments, and as explained in detail below, we have gathered more evidence in support of 'bet hedging' behaviour in the unicellular microalga *Haematococcus pluvialis* (that is distinct from the stress response, resulting in secondary carotenogenesis or carotenogenic response (CR)). Therefore, the chosen title, "Bet hedging in a unicellular microalga", is an objective description of the main result of the manuscript. Note that the 'bet hedging' behaviour we describe here is distinct from a stress response. Most importantly, the non-mobile green stage is always present – this is particularly noteworthy in the early stages of cultures, when there is no stress and the majority of the culture occurs in the mobile stage. We have therefore proposed an updated version of the life cycle (see Figure 1a), which we have modified in our revision to clarify our main points. We have also modified the

manuscript to make this more clear (lines 78-81, 349-353, 409-415, 876/877, 939). We apologise for any confusion we may have caused and hope that this point is now much clearer.

L20: ...adapt to unpredictable future... sounds a bit like an oxymoron (in a sense it is not really possible for a biological entity to adapt to a thing which did not yet happen). It is possible to prepare for this, to make some arrangements beforehand, but this is not an outcome of a factor which acts at present, it is more on the side of intelligent behavior which is not in the scope of this paper. On the other hand, it is quite possible to adapt to a volatile environment "assuming" that the environment will retain this characteristic in future... It is actually selection for the breadth of reaction norm, in terms of ecology and evolution.

Thank you for your suggestion. We agree with your point and have rephrased the sentence accordingly (lines 13-14): "Understanding how organisms have adapted to persist in unpredictable environments is a fundamental goal in biology.

Please consider and possibly rephrase: "Secondary carotenogenesis as a bet hedging strategy ... in microalgae..."??? Please see also comments and questions below.

As briefly indicated above, the carotenogenic response is not the focus of this study, but a phenomenon that results in a morphologically indistinguishable phenotype compared to the onset of the carotenogenic response – the non-mobile green stage – but the function of which is different. Although interesting, the carotenogenesis in response to stress in *H. pluvialis* has been well studied and described. Here, we focus on a behaviour that can be observed much earlier than that – and also a behaviour

that is not displayed by the entire population (but the minority) – a 'bet hedging' behaviour, based on the observation that a mobile green population is always differentiated into mobile and non-mobile green subpopulations, with the non-mobile subpopulation trading reproduction for stress resistance. Particularly in the early time points of culture, this is not a stress response but bet hedging behaviour and is different from the traditional sensing and responding to stress (such as CR). We therefore considered the proposed title but decided against it as it does not reflect the focus of the article.

Nevertheless, we acknowledge that *H. pluvialis* is a species capable of carotenogenic response, as evidenced by its ability to accumulate the secondary carotenoid astaxanthin, which manifests itself in a bright red colour. This phenomenon is described in lines 71/72, 409-428 and in the graphical overview (red cells) (Fig. 1a). It is generally believed that these red cells are the final resistant form of *H. pluvialis*, also known as red aplanospores, capable of surviving in extremely harsh conditions. In contrast to many other carotenogenic microalgae with a growth-devoting green phenotype and a stress-defence carotenoid accumulating phenotype, our model organism technically has three phenotypes, i.e., mobile green, non-mobile green, and red (carotenoid accumulating) phenotype.

Taken together, although CR is a very interesting topic, it is not relevant to this study. Perhaps these red cells in the graphical overview (Fig. 1a) are somewhat misleading, but the original intention of including red aplanospores there was to show where the bet-hedging behaviour fits into the complete life cycle of *H. pluvialis*. We apologise for

any confusion this may have caused and have made changes to Fig. 1a to highlight where the focus of this research lies.

L33: storing -> storing

Changed as suggested (Line 25).

L74: it is not a strain, it is a species.

Changed as suggested (Line 63).

Judging from Fig. 1, the increase in immotile cells takes place at the onset of stationary phase of growth (roughly, from the middle of the shown observation period, d 14). Before that, the proportion of different cell types change more or less synchronously. It is accepted that the preparation to the induction of carotenogenic response (including the formation of immotile palmelloid cells) starts when the cell division rate slows down, so the metabolic sink for the photosynthates shrinks. Effectively this situation (displacing the balance of the production of photosynthates and the metabolic demand) resembles the onset of a stress provoking the above-mentioned changes (loss of cell motility and, later on, haematocyst formation). Fig. 1D shows that the cell suspension possessed, at advanced stages of cultivation, a high density, so at this stage the growth was likely limited by light. This is corroborated by the presence of linear phase at the growth curve (Fig. 1B). So the loss of motility can be, at least in part, due to a different stress (low light).

Therefore, "bet hedging" may be another interpretation of these well-known changes in the population of carotenogenic microalgae.

First, we need to clarify that Fig. 1d is not an image of cells at advanced stages of a

growth experiment. It is intended to illustrate the difference in behaviour between non-mobile and mobile green cells, i.e., non-mobile cells cannot swim and sink to the bottom, which is visible to the naked eye. To achieve this, we separated and enriched different cell types in a flask to take an image. When they are mixed together, they cannot be clearly distinguished. To clarify this, we have made changes in the text (lines 141-144, 882-884).

Second, the essence of the bet hedging theory is that an isogenic population diversifies into different phenotypes with various fitness even under favourable conditions. In our case, the culture conditions are regarded to be favourable for the growth of *H. pluvialis* cells – in fact, they are identical to the ones used by other researchers to achieve fast growth rates and high biomass production. However, we have to admit that these favourable growth conditions in batch experiments will gradually deteriorate over time, such as lower availability and higher competition for nutrients, self-shading caused by high density, and the release and accumulation of unfavourable substances from dead cells etc. When confronting these circumstances (highly likely at the stationary phase) and in our culture starting at day 8, and becoming very apparent from day 14 onwards, it is reasonable to expect cells to transform from growth-dedicated activities to stress-resistant metabolisms.

However, the key finding of this research is the observation of the early emergence of non-mobile cells - so in our cultures before day 8 (Fig. 1b, c). During this time, we believe that there should be no external stresses as listed above. Traditional stress sensing and response (such as CR) cannot therefore explain the early appearance of

the non-mobile cells at the beginning of the culture period and cannot explain why only a small fraction of the population changes, while the majority remain as mobile. As a consequence, we cannot agree with the statement that "So the loss of motility can be, at least in part, due to a different stress (low light). Therefore, "bet hedging" may be another interpretation of these well-known changes in the population of carotenogenic microalgae." Although light limitation or self-shading may occur in a high-density population, it is unlikely to occur in an early population (before day 8).

Finally, from day 14 and onwards, you are correct that the transformation from mobile cells to non-mobile cells accelerated. At this relatively late stage of growth, we agree that the external stresses, as discussed above, occurred and led to the observed shift in the population, which was then soon dominated by the stress-resistant non-mobile state. To clarify this, we have made changes in the text (lines 317-328).

L122: what is "spatial stress" for microalgae?

We use the term 'spatial stress' according to Ruhe et al., 2013: "Spatial stress is a general term used to describe the stress from the crowdedness of a population, given the existence of contact-dependent growth inhibition in microbial populations." We have included the citation in the manuscript (line 107).

Considering the results summarised in Fig. 2: those provide a solid confirmation of well-know stress resilience of the immotile cells of the carotenogenic microalgae. By the way, did the cell populations further dissociate into motile and immotile cells under the "standard" conditions (Fig. 2A)?

Yes, populations tested under standard conditions dissociate into mobile and non-

mobile subpopulations. Under standard conditions, the phenotypic diversification of a mobile population is repeatedly observed. In contrast, the non-mobile population will maintain the non-mobile state (lines 174-177).

The bottom line is that this study shows the applicability of the theory of "bet hedging" to the populations of microalgal using the illustrious example of carotenogenic microalga *Haematococcus pluvialis*. The most conclusive results in this context are constituted by the observation of the population structure under favorable growth conditions and the experiments on rediversification of the cell types upon re-inoculation into the fresh medium. The other part of the study represents a recapitulation of what is already known on the stress physiology and biochemistry of *H. pluvialis* (see e.g. Boussiba 2000 and Recht 2014 etc.). It would much more to the point to use continuous culture under favorable conditions to show the persistence of the population (re-)differentiation, the limitations inevitable in the batch cultivation experiments complicate reaching this goal. Ultimately, a kind of bet hedging of this sort can be found in many organisms, it is not quite correct to say that it is something newly discovered and only in these microalgae. This phenomenon actually has an established name---secondary carotenogenesis. Therefore, I cannot agree with the statements in LL65-68. It is also recommendable to explore some other non-carotenogenic microalgae species to find out of the so-called bet hedging is associated with the induction of the carotenogenic response (and is actually an alternative designation of it) or it is a more universal phenomenon in microalgae.

First, we agree that another option to demonstrate "bet hedging" would be to use

continuous culture under favourable conditions. This would however only show a longer persistence of the bet hedging behaviour as compared to our experiments in batch culture (note that in batch culture the number of the non-mobile cells stayed fairly constant for about 7 days). This and the fact that we do not have access to continuous culture, such as a chemostat, to set up the experiments, we decided for another set of experiments to demonstrate the population dynamics in batch culture under secondary carotenogenesis induced by stress. To directly show that the reported early emergence of phenotypic diversification is different from the traditional stress response, we examined the performance of mobile populations exposed to external NaCl stress. We found distinctly different population dynamics between bet-hedging populations (in the absence of NaCl) and populations under stress (in the presence of NaCl) with regard to population growth and mobility-based population composition (Supplementary Fig. 2). Specifically, compared to normally growing populations in the absence of NaCl, all other populations exposed to NaCl gradients showed different levels of growth arrest and cell death (Supplementary Fig. 2a). In terms of population composition, in contrast to the more stable population composition of populations grown without external stress (approximately 9.4% of the population was non-mobile), relatively low stress (3 g/L NaCl) resulted in a 50:50 ratio of both phenotypes from day 1 onwards, and thus in a significantly higher proportion of non-mobile cells on day 7, while all cells exposed to even higher NaCl concentrations (6, 9, 18 g/L) lost mobility on day 2 (Supplementary Fig. 2b). These results show that the population performance, i.e., growth and population composition, in the absence of external stress differs from

population performance under stress. Thus, the traditional stress response paradigm cannot explain the early phenotypic diversification. Instead, we propose that this phenomenon is a bet-hedging behaviour. Additionally, MAPK genes, the upregulation of which are thought to play an essential role in response to stress (as stated in lines 270-280), in non-mobile cells showed a relatively down-regulated pattern compared to that in mobile cells (Fig. 4d), providing more evidence that the transformation of the non-mobile cells was unlikely caused by stress. Relevant changes in the manuscript are: lines 113-131, 317-340, 349-353, Supplementary Fig. 2.

Second, you are completely right that the bet hedging strategy can be found in many organisms, as we stated in the abstract and introduction (lines 14/15, 52-54). However, it should be noted that, to our knowledge, although sometimes proposed, there has not been direct experimental evidence for bet hedging in microalga. We think that our results presented in the manuscript can fill this gap.

Finally, as explained above, the carotenogenic response, although very interesting, is not relevant to our story. We agree that more species should be included to support the idea that bet hedging is universal in microalgae. While we do not claim that we show the universality of this phenomenon in microalga, it is very likely that this behaviour is more common than we currently know. Although this is important, it is not the main focus of our study, and there is a high probability that we will not be able to observe or detect phenotypic diversification (either morphological or physiological) under favourable conditions as we did in *H. pluvialis*. This difficulty is pervasive in microbial bet hedging research, which partly explains why bet hedging is often

discussed in review articles, but reports with experimental evidence are comparatively rare. As we agree that some degree of replication of the bet hedging behaviour – beyond the strain used – would be desirable, and as we want to make a general statement about the species *H. pluvialis*, rather than including other species, we investigated whether the observed phenotypic diversification is strain-specific. To this end, we performed the culture experiment under favourable conditions with three additional *H. pluvialis* strains. Indeed, we found that all strains tested showed similar patterns of phenotypic diversification (Supplementary Fig. 3), suggesting that our results were not strain-specific but a general behaviour of *H. pluvialis*. Relevant changes in the manuscript are: lines 132-138, 432-442, Supplementary Fig. 3.

References:

1. Ruhe, Zachary C. et al. Hayes. "Bacterial contact-dependent growth inhibition." *Trends in microbiology* 21.5 (2013): 230-237.

Response to Reviewer #2

As it has been discussed, phenotypic heterogeneity in green stage of *Haematococcus pluvialis* has been reported in previous studies, although an early appearance was observed in this study. The early presence of non-motile cells was explained by the authors as a bet hedging strategy to deal with unpredictable changes in environment. Variations in growth, responses to adverse environment, physiological profiles as well as transcriptome, were examined between populations with distinct compositions. Although a great deal of interesting data has been provided in the manuscript, they are largely descriptive. No sufficient mechanistic insights into bet hedging in microalgae were disclosed.

Thank you for the critical review that motivated us to clarify the exposition of our work. So far the phenotypic heterogeneity in the green stage of *Haematococcus pluvialis* has been described as a stress response. In contrast, this study is the first to report phenotypic heterogeneity in the absence of stress. In a set of manipulative experiments, we demonstrate that the observed heterogeneity is indeed bet hedging behaviour as both cell types trade growth/reproduction versus survival/stress resistance, which is independent of the environment. In the next step, we provide the mechanistic underpinning of the stress resistance of the non-mobile fraction of the population.

Also, in our revised manuscript, we added results from additional experiments, where we 1) characterise the behaviour of *H. pluvialis* populations under stressful conditions (Supplementary Fig. 2) and 2) go beyond the description of a single strain, as you suggested, and demonstrate bet hedging behaviour for another three strains of *H.*

pluvialis (Supplementary Fig. 3).

Hence, we have to politely push back the statement that our work is "largely descriptive", as what we present here is much more than just observations. While we do not provide the detailed mechanistic underpinning of bet hedging in this species, these details – while interesting – are not relevant to our message of reporting and characterising the first instance of bet hedging in a microalga. In the revised manuscript, we updated the exposition and the discussion of our study to make our aims clearer.

We agree in principle that it would be very interesting to investigate the mechanistic underpinnings of bet hedging in *H. pluvialis*. Nevertheless, as accepted in the field of bet hedging research (and as highlighted in a recent focus article on bet hedging strategies in microbial communities by Morawska et al (2022) in "WIREs Mechanisms of Disease"): "*Nonetheless, much more remains to be discovered, including the mechanisms of stochastic switching, which is the most challenging to follow.*" Thus, such a project is beyond the scope of the current study.

One reason for the obstacle in understanding the underlying mechanisms of such phenotypic diversification, i.e., the factor driving the diversification, is that in the case of bet hedging, the population is phenotypically heterogeneous. If all individuals of a population transformed into another phenotype, it would be relatively easier to investigate underlying mechanisms, such as determining the signalling molecule or examining differences in metabolic activities. Nevertheless, in bet-hedging populations, only a fraction of a population diversifies, whereas other individuals remain the same. Consequently, traditional methodological approaches are usually not sufficient to

investigate bet hedging. To date, the most popular explanation for the occurrence of bet hedging is stochastic initiation of gene transcription (Ratcliff et al., 2010, Lycus et al., 2018), but why and how also requires more evidence. In our case, the early diversification of phenotypes is a simple but important indicator of stochastic gene expression, which we also mention in the manuscript. Relevant changes in the manuscript are: lines 20/21, 113-138, 312-314, 317-332, 333-340, 349-353, 402-408, 433-443, Supplementary Fig. 2 & 3.

Additionally, is it possible the early appearance of the phenotypic heterogeneity in this study is a strain-specific phenomena?

Thank you for this question. We agree that evidence of bet hedging from more strains of *H. pluvialis* would further support our proposal. Accordingly, we performed another round of growth experiments under favourable conditions with the original and three additional *H. pluvialis* strains. Indeed, we found that phenotypic diversification is not unique to the focal strain used in this study but is a general phenomenon for this species (Supplementary Fig. 3). Relevant changes in the manuscript are: lines 132-138, 433-443, Supplementary Fig. 3.

References:

1. Morawska, L. P., Hernandez - Valdes, J. A., & Kuipers, O. P. (2022). Diversity of bet - hedging strategies in microbial communities—Recent cases and insights. *WIREs Mechanisms of Disease*, 14(2), e1544.
2. Ratcliff, William C., and R. Ford Denison. "Individual-level bet hedging in the bacterium *Sinorhizobium meliloti*." *Current Biology* 20.19 (2010): 1740-1744.

3. Lycus, Pawel, et al. "A bet-hedging strategy for denitrifying bacteria curtails their release of N₂O." *Proceedings of the National Academy of Sciences* 115.46 (2018): 11820-11825.

Response to reviewer #3

The manuscript submitted by Tang et al. presents an elegant ecophysiological study of a unicellular microalga that exhibits a bet hedging. The bet hedging in microorganisms is a fascinating survival strategy that secures populations' existence in fluctuating environments, and this is the first communication of such phenomenon in microalgae. *Haematococcus pluvialis* aside from being an important freshwater primary producer, too has important biotechnological applications. The findings presented by the authors are noteworthy and of interest to a wide audience, even though the mechanism behind the bet hedging phenomenon in *H. pluvialis* remains elusive.

I enjoyed reading the manuscript, it is well written. The data set presented is clear and neat, with sufficient replications and statistical analysis. The methodology used is sound. The work supports conclusions and stated hypotheses.

I have no major comments and I support the publication of the manuscript.

Thank you very much for your enthusiasm for our work and evaluation of the manuscript, we have modified the manuscript according to your suggestions.

Minor comments:

-although names above the genus level are not always italicised, I encourage italics for names of all taxa regardless of rank

We corrected the corresponding contexts (line 61/62).

-in methodology, I suggest using rcf instead of rpm, or provide details such as model of rotor and centrifuge

We have added the model information of the centrifuge used (lines 554, 574/575, 580, 583/584, 621).

-in transcriptomic analysis, the removal of genomic DNA is a crucial, yet tricky step, as a single DNase treatment is not always sufficient and requires checks for residual genomic DNA and eventual repeat. Has the RNA been tested for presence of residual gDNA?

All our samples for transcriptomics were directly sent to a commercial sequencing company, so we did not extract mRNA by ourselves. According to the protocols, total RNA was extracted, and genomic DNA was removed using DNase I, then the quality, purity and concentration of extracted RNA were determined respectively (see details in the Materials and Methods section, lines 626-629). We double-checked with the company, there were no extra steps taken to check for possible contamination of gDNA. However, since the model organism is eukaryotic, we do not think gDNA will interfere with the results. Because in the step of mRNA isolation from total RNA, only mRNA of eukaryotic cells has a polyA tail, with the polyA selection method by oligo(dT) beads, then only mRNA can be isolated from the total RNA (even with gDNA inside) for further analysis. We also have implemented this information into the transcriptomics method section (lines 634-636).

-please check and correct the reference list. Several papers are listed at least twice in the reference list (e.g. Baumont et al., Lycus et al., Philippi et al.)

We have corrected these errors and double-checked the reference list.

Reviewers' Comments:

Reviewer #1:

Remarks to the Author:

I had a pleasure reading the explanations and the answers of the authors to the questions elicited by the interesting concept they developed in their study. The authors have also done a fair job of amending their paper to make it more focused and convincing. Indeed, what they call "bet hedging" deserves a close attention of the readership of the journal and further research by the scientific community. I do not object publishing of this research in revised form.

Reviewer #2:

Remarks to the Author:

The authors have addressed all my concerns. There is no further comment from my side. I support the publication of the manuscript in the present form.

Reviewer #3:

Remarks to the Author:

I have no further comments.